# Carbon Dots-Based Logic Gates

**DOI:** 10.3390/nano11010232

**Published:** 2021-01-17

**Authors:** Shweta Pawar, Hamootal Duadi, Yafit Fleger, Dror Fixler

**Affiliations:** 1Faculty of Engineering and the Institute of Nanotechnology and Advanced Materials, Bar Ilan University, Ramat Gan 5290002, Israel; sppawar.shweta@gmail.com (S.P.); hamootal@gmail.com (H.D.); 2Bar-Ilan Institute of Nanotechnology & Advanced Materials (BINA), Bar Ilan University, Ramat Gan 5290002, Israel; Yafit.Fleger@biu.ac.il

**Keywords:** carbon dots (CDs), logic gates, nanodevices, molecular logic, PET, IFE, FRET

## Abstract

Carbon dots (CDs)-based logic gates are smart nanoprobes that can respond to various analytes such as metal cations, anions, amino acids, pesticides, antioxidants, etc. Most of these logic gates are based on fluorescence techniques because they are inexpensive, give an instant response, and highly sensitive. Computations based on molecular logic can lead to advancement in modern science. This review focuses on different logic functions based on the sensing abilities of CDs and their synthesis. We also discuss the sensing mechanism of these logic gates and bring different types of possible logic operations. This review envisions that CDs-based logic gates have a promising future in computing nanodevices. In addition, we cover the advancement in CDs-based logic gates with the focus of understanding the fundamentals of how CDs have the potential for performing various logic functions depending upon their different categories.

## 1. Introduction

In many fields of science, carbon dots (CDs) have grabbed significant attention due to their unique chemical and physical properties. These properties include nontoxicity, inexpensive synthesis, and excitation dependent fluorescence performance [1,2]. In 2004, the CDs were developed accidentally during the electrophoretic purification of carbon nanotubes (CNTs) by Xu and coworkers [3].

The photophysical, as well as chemical properties of CDs, can be tuned by changing their size and by doping with different heteroatoms such as nitrogen (N), oxygen (O), sulfur (S), and boron (B) [4]. Both the biological and electronic properties of CDs, such as water solubility and biocompatibility along with their electron donor and acceptor behavior, resulted in their application in the field of bioimaging, drug delivery, biosensors, optronics, catalysis, and sensors [5,6,7,8,9]. Initially, CDs were considered as an amorphous allotrope of carbon. From XRD analysis a broad diffraction pattern of sp^3^ carbon atoms inside CDs was demonstrated. Recently, the crystalline structure of CDs was studied by some researchers. These crystal structures can be classified as CDs having a graphitic crystalline core and others having non-graphitic crystallinity [10]. Regarding optical properties, CDs show a strong absorption band in the UV region in the range of 240–350 nm. The shorter wavelength bands, which are 240–280 nm, correspond to π–π* transitions, while the longer wavelength transition in the range of 350 nm is for n–π* transitions. In some cases, additional bands are observed because of the presence of multiple functional groups on the surface of the CDs. CDs possess excitation dependent emission properties, which make them distinct from other carbon allotropes. The surface composition of CDs was studied through the X-ray photoelectron spectroscopy (XPS) technique. The surface of CDs mostly consists of an oxygen-containing functional group such as −COOH, −OH, a nitrogen-containing group such as −NH_2_, CONH_2_, and other groups depending on the doping and functionalization of the CDs. The sensing ability of CDs imparted due to the presence of different functional groups on CDs surface made them suitable candidates for performing logic operations due to their ‘Turn-ON’ and ‘Turn-OFF’ response with analytes (Figure 1) [11]. Typically, the surface of CDs is rich in groups such as −COOH and −OH. This functional group gets a negative charge in the solution. Thus, the stable metal ion-CDs complex formation is favored in presence of positive ions due to electrostatic interaction. Metals ions such as Cu^2+^, Hg^2+^, Fe^3+^ when dispersed in CDs solution lead to the quenching of the CDs fluorescence. This is due to electron transfer from CDs to metal ions, preventing the radiative recombination of excitons. It is a well-established fact that CDs act as electron donors, which leads to their interaction with metal ions. CDs can also act as electron acceptor depending upon their surface structure. The charge transfer complex is formed between CDs and organic molecules or when CDs are adsorbed on the semiconductor surface. Apart from this, the pH of the solution changes the emission intensity of CDs. This is probably due to the changes in the charge state of the surface functional group. This also confirms that the surface state of the CDs plays an important role in their interaction with metal ions and the analytes.

Logic operations are performed by the molecules using one or more inputs and producing measurable output signals. Logic gates are the elementary building block for any digital system, where there is a certain relationship between the input and output signal based on a certain logic. The seven basic logic gates include AND, NOT, NOR, XOR, OR, NAND, and XNOR. Along with it, there are integrated logic gates, such as INHIBIT (INH) and IMPLICATION (IMP) circuits. These logic gates have several applications, such as AND gates being used for data transfer, while NAND gates are used in alarms and buzzers. They are used in circuits that involve processing and computation. In 1993 for the first time, Silva et al. explored the ability of the molecules to perform logic operations [12]. The logic output can be observed by the change in emission or absorption intensity or wavelength and establish the basis of logic gate operation via the known Boolean arithmetic function [13]. These molecular logic gates are superior to their semiconductor comparable as they can provide different information available to molecular logic as opposed to voltage information only [14]. The input can be physical (temperature [15], pressure [16], pH [17], light [18]), biochemical (enzymes [19,20] nucleotides [21]), and chemical (atomic [12], molecular [22]). For molecular computing, it is essential to design the logic gates using nano regime electronic circuits. Logic gates mimicking electronic circuits can be fabricated using nanomaterials such as fluorophores [23], graphene oxide dots [24], CDs [25], copper nanomaterials [26], silver nanocluster [27], and more.

CDs can be considered as the green carbonaceous nanomaterial possessing great potential to replace the conventional fluorophore [28] and silicon-based logic gates techniques [29]. Recently, molecular logic gates performing logic functions produced captivating research of miniaturization in the field of information technology [30]. CD-based logic gates show strong potential for optical sensing, leading to new avenues for future advancement of multidirectional memory devices. There are several reviews on CDs based on their forensic applications [31], synthesis [2], bioimaging [28], and sensing [6]. In addition, there are reviews on molecular logic gates based on their past, present, and future [32], fluorescent sensors [33], and biological logic gates [34]. To date, no review has been devoted to the combination of CDs and logic gates. This review is focused on the application of CDs in logic functions. It covers the latest advancement and development in the field of CDs-based logic gates, along with understanding the mechanism of CDs-based logic systems. Here we have categorized logic output based on their received output as single, combinational, sequential, and reversible. In a similar manner, we have divided the CDs used for logic function depending upon their synthesis as pristine, functionalized, doped, co-doped and other CDs complexes.

## 2. Logic Output

CDs based logic gates can be categorized based on their displayed logic output. For this review purpose, we have categorized the CDs based logic gates as single output, combinational output, sequential output, and reversible systems (Figure 2). 

### 2.1. Single Output

The single output is the basic building block of CDs based logic gates. It is important to establish a complex logic function based on these simple single-output logic operations. This type of system responds to multiple inputs to give a single response. Due to the simplicity of these systems, many researchers have used CDs as fluorophores for developing CDs based logic gates. Lin et al. have demonstrated different logic functions such as YES, OR, NOT, XOR, and IMP based on the sensing properties of CDs with metal cations and anions. These multiple logic gates are created through sequential metal ion association and anion dissociation process with CDs [35]. Zhao et al. constructed an AND logic gate with AgNPs nitrogen-doped CD nanocomposites without any chemical labeling and complex modification [36]. There are several reports that have constructed AND logic operation through the detection of different metal ions, such as Cr^6+^ [37], Hg^2+^ [38], Cu^2+^ [39], and Fe^3+^ [40]. Apart from AND, other logic operations such as the INHIBIT function were also fabricated through sensing of various metal ions through different amino acid derivatized CDs [41], histidine [42], Cu^2+^, H_2_S [43], arginine, and acetaminophen [44]. Other researchers developed IMPLICATION logic operations using the sensing ability of CDs for Hg (II) and cysteine [45], AA [46], Fe^3+^ [47], Hg^2+^, and biothiols [48], Hg(II), and glutathione [49]. A multilevel single-output logic system was also developed using gadolinium doped CDs with H^+^, OH^−^, Cu^2+^ as inputs, which trigger both Fluorescence intensity (FI) and magnetic resonance (MR) signals [50]. Apart from the fluorescence technique, using magnetic resonance signal for dual readout logic operations is of significant importance, as the combination of FL/MR techniques gives the logic devices better applicability in case of biological application.

### 2.2. Combinational Logic Output

The output of the combinational logic operation is the instant response to their current input state as logic 0 or logic 1. This type of output depends upon the combination of the input all the time. Thus, the combinational logic circuit is termed as ‘memoryless’. Combination logic circuits combine or connect simple logic operations to build a complex logic circuit. Tang et al. demonstrated the combinational nano logic gate with a dual output channel. The supramolecular assembly based on CDs showed two distinctive patterns of logic function at two different emission wavelengths of 440 nm and 490/545 nm. The output channel at 490/545 nm consists of a combination of two INHIBIT gates [51]. The supramolecular strategy serves as a substitute for covalent modification and simplifies the fabrication process. Zhao et al. performed half addition and half subtraction operations on synthesized pH-responsive CDs at two different emission wavelengths. A half-adder was constructed by combining XOR and AND gates, which further implement the function of sum and carry, while the half-subtractor consisted of the INHIBIT gate producing borrow bits and XOR gate for obtaining difference bits [52]. Fan et al. have designed a three-input and three-output combinational circuit along with a keypad lock using red emissive CDs/Prussian blue composite electrode films. The complicated logic gate was constructed using elementary functions such as OR, AND, INHIBIT, and IMP [53].

### 2.3. Sequential Output

Unlike combinational output, the third category is the sequential output in which the output depends on both present inputs and previous output. In contrast to combinational output, it has a memory, so the output varies based on the input. Qu et al. designed multiple single and sequential DNA-based logic gates. These types of logic gates were inspired by B to Z-DNA transition induced by functionalized CDs. The logic gate was constructed based on FRET between CDs and DNA intercalators and fluorescence quencher for CDs. Single AND functions were established at 585 nm and NAND logic at 465 nm. Similarly, AND + INHIBIT and NAND + INHIBIT sequential circuits were constructed at 585 nm and 465 nm, respectively [54]. Fluorescence techniques have certain limitations, such as shorter emission lifetime of nanoseconds leading to inner filter emission (IFE), overlapping of excitation and emission spectra, interference from the light scattering, and short-lived autofluorescence species. These limitations can be overcome by the triplet excited state phenomena known as ‘Phosphorescence’. Wang et al. developed a phosphorescence-based OR-INHIBIT logic gate using inputs such as Hg(II), tDNA (target ssDNA), and doxorubicin [55]. The phosphorescence logic gates are superior to most common fluorescence-based logic gates due to their benefits over fluorescence. Viswanathan et al. used the switching nature of CDs to design memory devices having sequential circuits due to reversible response with the addition of Hg(II) and L-cysteine alternatively. A Write-Read-Erase-Read nature of sequential circuit was developed using OFF-ON reversible behavior with inputs as Hg (II) and L-cysteine. [56]. Other research groups have reported integrative logic gates such as NOR and INHIBIT (INH) and IMPLICATION (IMP), NOR and AND logic functions. These logic gates have three inputs such as Zn(II), pH 2, and Cu(II) for NOR and INH and four inputs such as Zn^2+^, S^2-^, Cu^2+^, and pH 2 for IMP, NOR, and AND logic operation [57]. Recently, an integrative logic system based on dual readout logic devices with both magnetic resonance (MR) and FI of holmium doped CDs was discussed by Fang et al. The multi-readout logic circuits were developed by recording the same signal by two different readout techniques. The chemical inputs were H^+^, Fe^3+,^ and Fe^2+^, while the fluorescence output was recorded at 440 nm along with magnetic intensity. A fluorescence-based NOR-INHIBIT and MR-based (XOR-INHIBIT)-OR sequential logic system was demonstrated successfully [58].

### 2.4. Reversible Output

The reversible system is interesting because it allows the reassessment of the output [59]. The advantage of this process is that the logic operation can be performed multiple times without adding more analytes constantly. The reversible system with the activated condition can revert to its original state with the introduction of a reactivator. CDs-based reversible logic gates work on the principle of quenching the FI signal, which further recovers by a recovery agent [43]. This type of logic gate is a potential candidate for low-power computing [60].

## 3. Sensing Mechanisms of CDs Based Logic System

The majority of the CDs based logic gates depend upon the changes in the fluorescence response of CDs. These changes either in FI or wavelength make them potential candidates for performing various logic operations, such as AND, OR, NOR, INHIBIT, IMPLICATION, etc. [32]. Different sensing mechanisms are responsible for inducing the fluorescence changes of the CDs based logic system. Initially, CDs interact with the quencher leading to the OFF state, then with the introduction of the recovery agent, the FI of CDs is recovered bringing it back to the ON state. These interactions can be noncovalent, which includes hydrogen bonding, π–π interactions, donor and acceptor, co-ordination based, or electrostatic, and covalent interactions [61]. When CDs interact with the quencher molecule, the FI is quenched due to nonradiative energy transfer from the donor to the acceptor. A different mechanism, such as PET, FRET, and IFE, is responsible for the energy transfer process as depicted in Figure 3. In the case of PET mechanism, there is a redox reaction upon irradiation between CDs (donor) and the other analytes (acceptor), which can donate and accept electrons, leading to the formation of a non-fluorescent complex. In the case of FRET, the excited CDs while returning to the ground state transfers their energy non-radiatively to the acceptor molecule. Theoretically, the rate of energy transfer depends on: (a) The orientation of the donor and acceptor, (b) the extent of overlap between the donor emission and acceptor absorption spectrum, and (c) the separation distance between the donor and acceptor that should be less than 10 nm.

Between static and dynamic quenching, dynamic quenching dominates the FRET quenching process as energy transfer takes place in an excited state [62]. Another mechanism of FI quenching is IFE, primarily IFE due to excitation beam attenuation in highly concentrated samples. In this case, the fluorophore facing the excitation fluoresce brighter than the ones at the center, which strongly affects the detected signal. In secondary IFE, there is a significant overlap between excitation and emission spectrum, and the emitted light is reabsorbed by the sample, leading to the quenching of the FI signal. Most of the CDs based logic gates operate on the mechanism of efficient FI quenching of CDs due to the above-mentioned phenomena, and the occurrence of restoring agents brings back the fluorescence effectively leading to the sensing of various analytes. The sensing mechanisms of CDs based logic systems are shown in Figure 3.

## 4. Carbon Dots Design for Logic Function

The design of CDs for performing logic functions is generally based on five categories (Figure 4): Pristine CDs, functionalized CDs, doped CDs, co-doped CDs, and other complexes with CDs.

### 4.1. Pristine CDs

The first category of design based on the synthesis procedure is the direct interaction of CDs with the analytes in their pristine form. The interaction is possible due to the presence of the different functional groups on the CDs surface. Hu et al. have used the as-prepared CDs for the determination of auramine O. In their work instead of fluorescence intensity (FI), the appearance of two peaks was taken as an output. They demonstrated dual emission, which is the auramine O-stimulated response [63]. Chattopadhyay and co-workers have synthesized CDs for logic operation in two different phases, i.e., solid and liquid phases. The direct interactions of CDs with organic molecules and metal ions in both phases achieved the basic and integrated logic operations [64]. This kind of operative multiphase logic response are scarce in literature, developing a solid phase integrated logic function could be a good option. In the year 2016, Hu et al. prepared a D- Penicillamine sensing fluorescence switch sensor whose logic gate function was based on the luminescent recovery of CDs (Figure 5). The quenching of FI was done through Hg^2+^ ions, which was retrieved with the addition of D- Penicillamine [65].

The following year, in 2017, Hu et al. designed CDs for sensing glyphosate (Gly) based on fluorescence resonance energy transfer (FRET). The logic function is based on FI quenching of CDs in presence of Gly due to energy transfer between the donor (CDs) and acceptor (Gly) [66]. In the same year, two other research groups, Zhao et al. and Dong et al., used pristine CDs for intracellular pH sensing, and arginine and Cu^2+^ detection [11], respectively. A three-state switch was obtained by controlling the fluorescence emission at different pH levels [52]. Later, Vishwanathan et al. derived CDs from pineapple peels. Logic operations, such as implication (IMP) and NOT gate, were generated using Hg^2+^ and cysteine as inputs [56]. Chromium (Cr^6+^) and cysteine were detected using red emissive CDs via dual modes, such as colorimetry and fluorescence by Dong et al. The red-emitting CDs constructed an AND logic gate [37]. A dual signal sensing of Hg^2+^ and glutathione was achieved by down and upconversion CDs with high quantum yield (QY) −62%. The logic operation was obtained through the “ON-OFF-ON” process [38]. Recently, Shuang et al. prepared blue CDs that are excitation independent and used them for Fe^3+^ and F^-^ sensing. An AND molecular logic gate was devised with the help of Fe^3+^ and F^-^ as chemical inputs [40].

### 4.2. Functionalized CDs

The second category includes the functionalization of the CDs. The functionalization imparts specific molecular recognization abilities to the CDs depending upon the organic molecule used to functionalize the surface. Qu et al. for the first time reported spermine-functionalized CDs (SC-dots) to induce right-handed B DNA to left-handed Z DNA transition under physiological conditions. Furthermore, a variety of logic gates were constructed based on FRET between fluorescent CDs and DNA intercalators, such as Ethidium bromide (EB) and FI quenching of CDs with iodide ions [54]. Namasivayam Dhenadhayalan and King-Chuen Lin prepared two types of CDs, COOH functionalized CDs and amine-functionalized CDs. The significant observation of their study was that the functional group present on the CDs surface plays an essential role in both cation and anion detection. In addition, the interaction of these functional groups with the ions and anions leads to the construction of different types of logic gates, as shown in (Figure 6) [35]. This study confirms that the functional groups present on CDs surfaces imparts certain association affinity of CDs towards certain metal ions. Xia et al. developed phosphorescence-based logic gates using surface-modified CDs, ssDNA, and graphene oxide (GO). Room temperature phosphorescence (RTP) based logic operations, such as INHIBIT, OR, and OR-INHIBIT were designed using inputs such as Hg^2+^, DOX, and tDNA and phosphorescence-ON as an output signal [55].

For the limited tunability of the CDs spectrum, it is essential to improve the homogeneity of local electronic structure by controlling the size of sp2 domains, the surface functional group through surface passivation. Thus, the electronic properties of CDs were systematically modulated with ethylenediamine and different amino acids, such as cysteine, lysine, histidine, and arginine by Kuei and group. This kind of derivatization resulted in tuning the selectivity of sensing metal cations. The logic gates obtained from the fluorescence response of CDs depend on the fluorescence quenching in presence of metal ions [41]. Das et al. surface quaternized the CDs with benzalkonium chloride, which was synthesized from seaweed and lemon juice abbreviated as KLBC-dots. The functionalized probe acts both as a bifunctional fluorescent sensor for Cr^6+^ and ascorbic acid (AA), and also as an antibacterial agent. The logic operation was based on the inner filter effect (IFE) [67]. In another study, Zhang et al. capped CDs with polyethyleneimine and used them bioassay of Cu^2+^ and S^2^. The INHIBIT Boolean logic is influenced by the FI quenching and recovery process in presence of analytes [43].

### 4.3. Doped CDs

The third type of design is based on doping. The electronic characteristic of the CDs can be effectively adjusted with chemical doping, which further improves their optical properties [68]. Zhao et al. reported the green synthesis of nitrogen-doped CDs and silver nanoparticles composites (AgNPs/N-CDs), with surface-enhanced Raman scattering (SERS) properties. A simple AND logic system was obtained based on AgNO_3_ (silver nitrate) and NaOH (sodium hydroxide) as inputs. The output signal was the presence of an absorption peak at 405 nm due to the formation of AgNPs/N-CDs nanocomposites [36]. There are very few reports where the absorption is used as an output signal for logic operation. Most of the available CDs showing logic gate application emits in the blue region, however, it is better to shift to a longer wavelength, which increases the penetration depth and is harmless to tissues. Dong and co-workers used bright green emissive nitrogen-doped CDs for Fe^3+^ and AA sensing, cellular imaging, and logic gate applications. The FI of doped CDs was quenched with the addition of Fe^3+^ due to photoinduced electron transfer (PET), which was further recovered by AA addition [69]. In another study by Chen’s group, they have synthesized three different types of nitrogen-doped CDs using citric acid as a source and small molecules such as ethanediol, ethanolamine, and ethidene diamine as the dopant. They were used for Hg^2+^ and biothiol sensing and further an IMPLICATION logic gate was created [48]. The N doped CDs were also used for sensing Cu^2+^ and GSH and an AND logic gate was constructed based on their sensing behavior [39]. Not only doped CDs were used for metal ion, biothiols, ascorbic acid detection. In the next research, Yang et al. used doped CDs for recognition of a phenothiazine drug known as chlorpromazine hydrochloride (CPH). An AND logic gate was constructed using N-CDs and CPH as inputs and quenching of FI as an output (Figure 7) [70].

For better feasibility and biomedical application, it is important to investigate dual readout logic devices. Yi et al. synthesized holmium (Ho^3+^) doped CDs (Ho-CDs), which exhibited pH responsiveness for both FI and magnetic resonance (MR) signals. In this study, the inputs were H^+^, Fe^2+^, or Fe^3+^, and the change in FI and MR signals served as an output for multilevel logic operations [58].

### 4.4. Co-Doped CDs

Co-doping is another important strategy for tuning the dopant population, magnetic properties, and electronic properties. It is used to enhance the solubility of CDs along with improving their stability and optical properties [71]. Li et al. developed novel magnesium and nitrogen co-doped CDs for selective sensing of Hg^2+^ and cysteine via FI quenching due to the formation of a non-fluorescent complex of CDs and Hg^2+^, and the recovery was done with the addition of cysteine. An IMP logic gate was constructed using Hg^2+^ and Cys as chemical inputs, and a change in the FI of Mg-N-CDs as an output signal [45]. Nitrogen and sulfur co-doping plays a vital role in enhancing the QY of CDs, as nitrogen and carbon have almost the same size which makes nitrogen doping easy and feasible. While sulfur tunes the fluorescence emission toward a longer wavelength by providing emissive traps for excited electrons. The work reported by Guttena’s group emphasizes converting toxic cigarette butts into nontoxic fluorescent N, S (nitrogen, Sulphur) co-doped CDs. In this work, the waste cigarette butts were converted to fluorescent CDs by a simple hydrothermal process without using any expensive instruments and having a variety of application. The synthesized CDs used for the detection of Fe^3+^ and AA, based on their sensing characteristic an IMPLICATION logic gate, was constructed. Apart from this, these co-doped CDs were used as invisible ink for security applications [47]. In another study, N-S doped CDs were used for sensing Cr^6+^ and AA based on the IFE and redox reaction. The fluorescence behavior of co-doped CDs was used to design molecular logic gate operations [72]. Mobin et al. synthesized the N, S co-doped CDs (N_S@RCD) from green alternatives such as Rosa indica, and use it for Au^3+^ and S^2-^ detection. This was the first report in which CDs act as a turn-on sensor for S^2-^ without the formation of any intermediate quencher complex. The system exhibits a single input ‘YES’ and multilevel INHIBIT logic functions (Figure 8) [73].

### 4.5. Other Complexes with CDs

Blue and green emission of CDs limits their use in biological applications as the penetration depth is low and autofluorescence is maximum. Therefore, it is important to have CDs that emit towards a longer wavelength. To achieve this goal Song et al. fabricated silver-CDs nanohybrid with infrared fluorescence. The FI signal was enhanced with metal enhanced fluorescence. The nanohybrid was used to detect AA, an antioxidant. An IMP logic gate was constructed using Fe^3+^ and AA [46]. Satnami et al. used the aggregation and dispersion of gold nanoparticles (AuNPs) in the presence of CDs to sense pesticides. A FRET-based logic gate was designed using toxic pesticides as inputs. A combination circuit consisting of INHIBIT and OR circuit was developed [25]. Similarly, Fang et al. used AuNPs and CDs hybrid for sensing fluoride ions using a bridge of 3-mercapto-l,2-propanediol. An AND molecular logic gate was developed based on anions and metal ions as inputs [74]. External factors, such as temperature, concentration, and solvent, affect the emission of the fluorescence sensing system. In order to overcome these limitations, researchers shifted their focus to ratiometric sensing, as it depends upon the calibration of two peaks and avoids the influence of the external and instrumental factors. Singh et al. used CDs and rhodamine-based ratiometric complex for the detection of histidine. The sensing systems depend on the FRET between CDs and rhodamine derivatives. An INHIBIT logic gate was implemented using Fe^3+^ and histidine as the inputs [74]. Later, Gui et al. used CDs and DNA template copper nanoclusters for ratiometric sensing of arginine and acetaminophen. Based on the FI response, an INHIBIT logic gate was designed [44].

## 5. Conclusions

In the present review, a comprehensive study was made on the advancement of CDs-based logic gates during the year 2013–2020. It consists of their synthesis, sensing mechanism, different types of possible logic operation, and we have also discussed the design of the CD sensor for logic operation based on the nature of CDs. This review can help researchers develop more low cost and biocompatible CDs-based chemosensors for logic gate applications. Developing molecular logic devices with small molecules or biomolecules gives a huge impact on modern-day science. With the current pandemic situation, the development of smart medicine will be very helpful. There are some reports available that discuss one of the strategies to fight the outbreak of COVID 19 is the real-time monitoring of proteolytic activity with potential protease inhibitors as biosensors for COVID 19 [75]. In one of the reports, COVID 19 genome was used as the input for an AND logic function. Two types of fragments were used as inputs. This detection was based on exonuclease III and DNAzyme [76]. Thus, CDs based logic gates can act as smart materials that can respond to various analytes resulting in the development of smart nanodevices. Table 1 summarizes most of the logic systems discussed in the text ordered by their year of publication.

## 6. Future Perspectives

CDs-based logic gates serve as potential agents for sensing various analytes, and thus can be used to design smart biocompatible nanodevices in the future. With the significant advancement in computational power, it will be interesting to develop computation in human bodies using these CDs-based logic systems. Fluorescent lifetime (FLT) may provide great applications for CDs-based logic gates owing to its advantages over steady-state FI detection [78,79,80]. This phenomenon is concentration and excitation quality independent. Other factors, such as sample thickness, absorption by sample, FI, and photobleaching also do not affect FLT measurements. FLT imaging microscopy (FLIM) is another imaging technique that can be effectively used to monitor the biologically driven logic gates. Most of the CDs based logic gates outputs are based on FI technique that can be replaced by phosphorescence as it has many key merits, such as longer emission FLT, minimum interference from scattered light, and short-lived autofluorescence species, and a wide gap between emission and excitation spectra. Though CDs are good replacements for toxic quantum dots and silicon-based technology, low QY and blue emission restrict their applications in biological applications. Therefore, it is important to take measures to increase the QY and to shift the spectra towards longer wavelengths, as the penetration depth of electromagnetic radiation is higher and the autofluorescence of tissue is minimum at the near infra-red window. There are very few reports on red-emitting CDs based logic gates. Recently, Gadolinium-doped [81] and holmium-doped CDs were used as contrast agents for generating MRI signals, to construct the dual emission mode CDs based logic gates. Both of them are unsuitable for biological applications due to the toxicity issues, hence it is essential to synthesize a biocompatible contrast agent.

## Figures and Tables

**Figure 1 nanomaterials-11-00232-f001:**
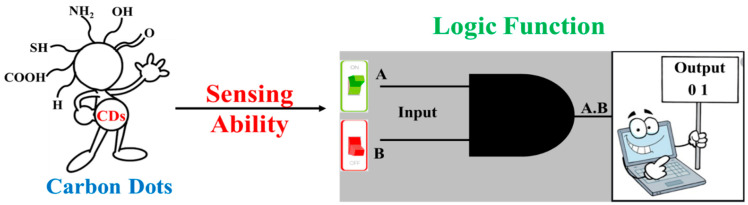
Carbon dots and their application in logic function based on their sensing ability.

**Figure 2 nanomaterials-11-00232-f002:**
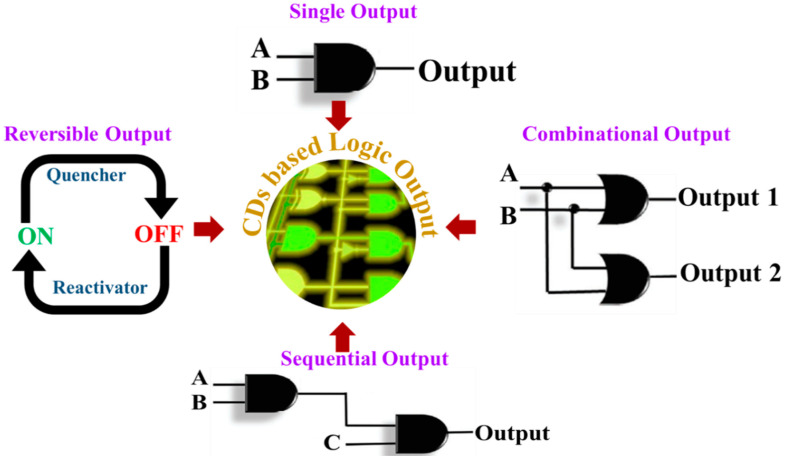
Different categories of carbon dots (CDs)-based logic gates according to their output. I. Single output, where only one output is generated. II. Combinational outputs are the integration of simple logic operations to obtain the complex combinational output. III. Sequential output, which responds to multiple inputs but with different stages of activation that should happen in a predestined order. IV. Reversible systems can switch between ON and OFF states depending upon the input added to the system.

**Figure 3 nanomaterials-11-00232-f003:**
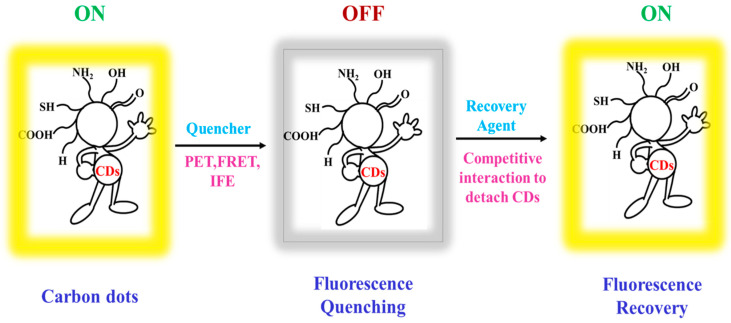
Mechanism of sensing CDs based logic systems with photoinduced electron transfer (PET), fluorescence resonance energy transfer (FRET), and inner filter effect (IFE) process for quenching and recovery by interactions with recovery agents.

**Figure 4 nanomaterials-11-00232-f004:**
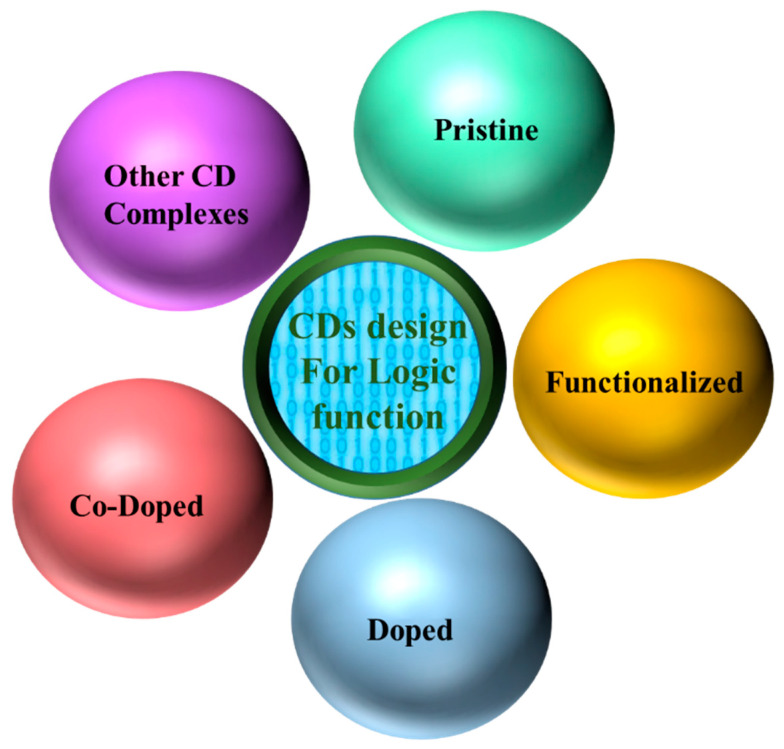
Five different categories of CDs design for logic function.

**Figure 5 nanomaterials-11-00232-f005:**
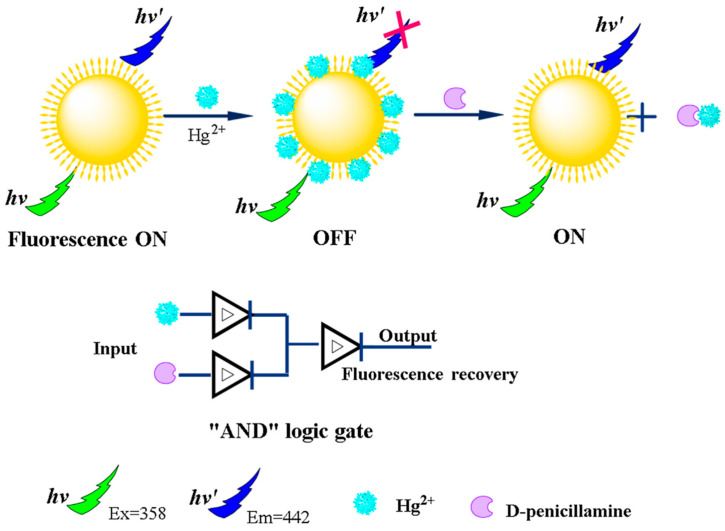
Schematic diagram of D-Penicillamine (D-PA) detection by pristine CDs and its AND logic gate function. Reproduced from ref [65] with permission of Sensors and Actuators, B: Chemical copyright 2016.

**Figure 6 nanomaterials-11-00232-f006:**
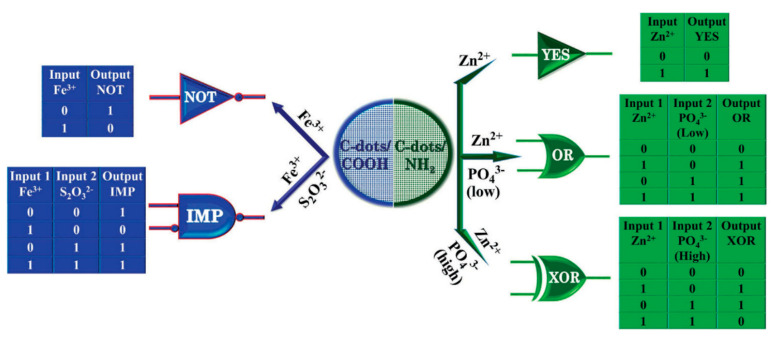
Functionalized CDs (COOH and amine functionalization) used for the detection of Fe^3+^, Zn^2+^ cations, and S_2_O_3_^2-^, PO_4_^3-^ anions with its multiple logic operations. Reproduced from ref [35] with permission of Scientific Reports copyright 2015.

**Figure 7 nanomaterials-11-00232-f007:**
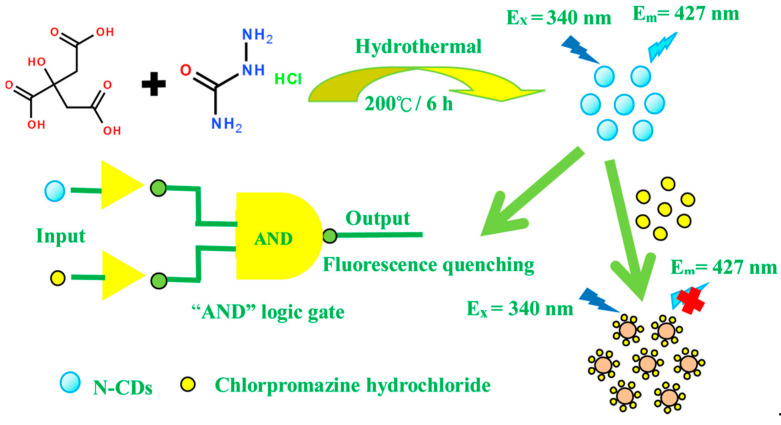
Schematic diagram of the formation process of N-CDs and detection for chlorpromazine hydrochloride (CPH). Reproduced from ref [70] with permission of the Journal of Photochemistry and Photobiology A: Chemistry copyright 2019.

**Figure 8 nanomaterials-11-00232-f008:**
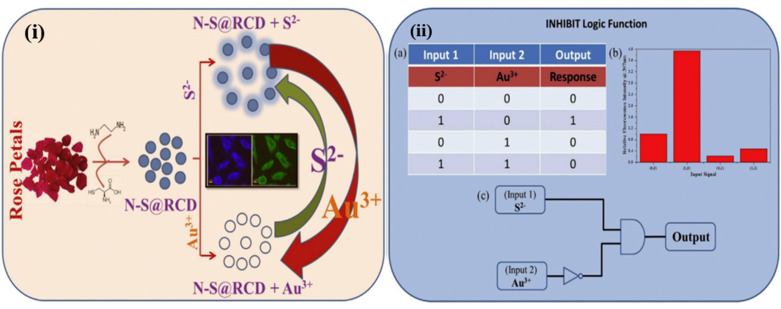
(**i**) Schematic illustration of multifunctional N-S@RCD. (**ii**) “INHIBIT” logic function using N-S@RCD. (**a**) Truth table, (**b**) fluorescence response of N-S@RCD under different inputs, and (**c**) Symbol of INHIBIT logic. Reproduced from ref [73] with permission of the Carbon copyright 2018.

**Table 1 nanomaterials-11-00232-t001:** Summary of CDs-based logic systems.

Types of CDs	Application	Logic Function	Type of Logic Output	Ref.
Spermine functionalized CDs	B to Z DNA transition	AND, NAND, AND + INH, NAND + INH	Single and sequential	[54]
Pristine CDs	O auramine detection	NOR-AND	Sequential	[63]
Pristine CDs	Metal ions (Fe^2+^, Fe^3+^) and organic molecules (Picric acid and H_2_O_2_) detection	NOT, OR, AND, NOR, NAND, NOT-NAND	Single and Integrated	[64]
Acid and amine-functionalized CDs	Cations and anions sensing	YES, OR, XOR, NOT, IMP	Single	[35]
Nucleic acid functionalized carbon dots	Phosphoresce logic gates were developed using cd, DNA, hg2+, dox	OR, INHIBIT, OR-INHIBIT	Single and sequential	[55]
Surface quaternized cationic CDs	Phosphate detection	YES, two INHIBIT	Combination	[51]
Silver Nanoparticles/N-Doped CDs Nanocomposites	SERS	AND	Single	[36]
magnesium and nitrogen co-doped CDs	Hg^2+^ and cys detection	IMPLICATION	Single	[45]
Pristine	D-Penicillamine detection	AND	Single	[65]
Silver-CDs nanohybrid	AA detection	IMPLICATION	Single	[46]
Pristine CDs	Glyphosate detection	AND	Single	[66]
CDs-MnO2 adduct	NAC detection	YES	Single	[77]
Pristine CDs	Intracellular pH sensing	XOR-AND, INHIBIT-XOR	Combination	[52]
Pristine CDs	Arginine and Cu^2+^ detection	AND	Single	[11]
Amino acid derivatized CDs	Detection of Pb, Hg^2+^, Fe^3+^, Zn^2+^, Cr^3+^, Cu^2+^	AND, INHIBIT	Single	[41]
Nitrogen doped CDs	Fe^3+^ and AA detection	AND	Single	[69]
Pristine CDs	fluoride ions detection	NOT, IMP, NOT-AND-OR	Single, sequential	[56]
Pristine CDs	Cr^6+^ and Cys detection	AND	Single	[37]
Polyamine coated CDs	Zn^2+^, Cu^2+^, S^2-^ and H^+^ detection	IMP-NOR-AND, NOR-INH	Integrative	[57]
N, S-Codoped CDs	Fluorescent Film, Security Ink, Bioimaging, Fe^2+^, Fe^3+,^ and AA Sensing	IMP	Single	[47]
Pristine CDs	Hg^2+^ and glutathione detection	AND	Single	[38]
N,S- Codoped CDs	Cr^6+^ and AA detection	AND	Single	[72]
N,S- Codoped CDs	Au^3+^ and S^2−^detection	AND	Single	[73]
Nitrogen-doped CDs	Hg^2+^ and biothiols Detection	IMP	Single	[48]
Benzalkonium chloride functionalized CDs	Cr^6+^ sensing and antibacterial activity	AND	Single	[67]
N-doped CDs	Hg^2+^ and glutathione detection, cell imaging	IMP	Single	[49]
Polyethyleneimine-capped fluorescent CDs	Cu^2+^ and H_2_S detection	INHIBIT	Single	[43]
CDs-gold nanoparticle	Detection of pesticides	INHIBIT-OR	Combination	[25]
Nitrogen-doped CDs	Cu^2+^ and GSH detection	AND	Single	[39]
Nitrogen-doped CDs	chlorpromazine hydrochloride detection	AND	Single	[70]
Pristine CDs	Fe^3+^ anf F^-^ sensing	AND	Single	[40]
Holium doped CDs	dual imaging	XOR+INH-OR	Integrative	[58]

## Data Availability

All data are available upon email request. Restrictions apply to the availability of these data. Some data are not publicly available since some articles are not open access.

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
