# Peer review of "Carbon Dots-Based Logic Gates"

_nanomaterials, 2021, doi:10.3390/nano11010232_

Round 1

Reviewer 1 Report

Pawar and co-workers have submitted a review about the potential applications of carbon dots (C-dots) for smart nanoprobes with the title “Carbon dots based logic gates”.

This manuscript presents a survey on C-dots as logic gates based on their (optical) response to the interaction with particular analytes, ions, and so on. In particular, the authors intend to gather the attention on particular classes of carbon dots materials as suitable systems in computing nanodevices and information processing, especially for biomedical applications.

The argumentations are convincing and support the emerging role and the outstanding characteristics of the nanomaterials considered in this review. So, I recommend this manuscript to publish on Nanomaterials after major revisions.

Comments and suggestions:

  • The review should be extensively revised by improving the section “ Carbon dots design for logic function”. In particular, a large variety of C-dots are reported but little information on the characteristics can be found, i.e. crystal structures, optical properties, surface properties. Moreover, this reviewer highly recommends explaining in depth the origin of interaction with analytes and ions.
  • Furthermore, it is not clear the state of the art of logic gates applications. Is it already employed in current applications? Is there a fundamental difference with the sensing application?
  • In “Doped CDs” section the authors do not justify the sentence “it is better to shift to a longer wavelength”. On the contrary, the necessity of high wavelengths emissions is well explained in the next paragraph.
  • The authors should illustrate the sensing mechanism in the first paragraphs of this review, in order to make easier the readability.
  • The reference to the pandemic problem of Covid-19 in the conclusions deserves more space in the review.

Author Response

We would like to thank the reviewer for their detailed and constructive comments, which have greatly improved the manuscript.

Reviewer #1:

This manuscript presents a survey on C-dots as logic gates based on their (optical) response to the interaction with particular analytes, ions, and so on. In particular, the authors intend to gather the attention on particular classes of carbon dots materials as suitable systems in computing nanodevices and information processing, especially for biomedical applications. The argumentations are convincing and support the emerging role and the outstanding characteristics of the nanomaterials considered in this review. So, I recommend this manuscript to publish on Nanomaterials after major revisions.

The review should be extensively revised by improving the section “Carbon dots design for logic function”. In particular, a large variety of C-dots are reported but little information on the characteristics can be found, i.e. crystal structures, optical properties, surface properties.

Answer: We thank the reviewer for his comment, and we are happy to elaborate on this issue. The following information on the characteristics of CDs was added to the introduction section (section 1) of the manuscript (lines 39-50 on pages 1-2).

Crystal structure: Initially CDs were considered as an amorphous allotrope of carbon. From XRD analysis a broad diffraction pattern of sp3 carbon atoms inside CDs was demonstrated. Recently, the crystalline structure of CDs is studied by some researchers. These crystal structures can be classified as CDs having a graphitic crystalline core and others having non- graphitic crystallinity. In the case of optical properties, CDs show a strong absorption band in the UV-visible region in the range of 240-350nm. The shorter wavelength bands which are 240-280nm corresponds to π-π* transitions, while the longer wavelength transition in the range of 350nm is for n-π* transitions. In some cases, additional bands are observed because of the presence of multiple functional groups on the surface of CDs. CDs possess excitation dependent emission properties, which makes them distinct from other carbon allotropes.

The surface composition of CDs is studied through the X-ray photoelectron spectroscopy (XPS) technique. The surface of CDs mostly consists of an oxygen-containing functional group such as –COOH, -OH, a nitrogen-containing group such as –NH2, CONH2, and other groups depending upon the doping and functionalization of CDs.

Moreover, this reviewer highly recommends explaining in depth the origin of interaction with analytes and ions.

Answer: We thank the reviewer for his comment. The following information on the origin of interaction with analytes and ions was added to the Introduction section (section 1) of the manuscript (lines 51-61 on page 2).

Mostly the surface of CDs is rich in oxygen-containing groups, such as –COOH and –OH. This functional group gets a negative charge in the solution. Thus, the stable metal ion-CDs complex formation is favored in presence of positive ions due to electrostatic interaction. Metals ions ,such as Cu2+, Hg2+, Fe3+, when dispersed in CDs solution lead to the quenching of the CDs fluorescence. This is due to electron transfer from CDs to metal ions, preventing the radiative recombination of excitons. It is a well-established fact that CDs act as electron donors, which leads to their interaction with metal ions. CDs can also act as electron acceptors depending upon their surface structure. The charge transfer complex is formed between CDs and organic molecules, or when CDs are adsorbed on the semiconductor surface. Apart from this, changing the pH of the solution changes the emission intensity of CDs. This is probably due to the changes in the charge state of the surface functional group. This also confirms that the surface state of the CDs plays an important role in their interaction with metal ions and the analytes.

Furthermore, it is not clear the state of the art of logic gates applications. Is it already employed in current applications? Is there a fundamental difference with the sensing application?

Answer: We thank the reviewer for his comment, and we are happy to discuss this.

Logic gates can be implemented with either optics, electronic, mechanical, or biological devices. They can be composed into the physical models of conceivable computation or algorithm. The sensing application of logic gates reviewed mostly uses fluorescence signals for their logic output. Other techniques use absorption, phosphorescence and magnetic resonance signals to derive molecular logic functions. Fundamentally different sensing techniques leads to various logic operation depending upon the probe used.

In “Doped CDs” section the authors do not justify the sentence “it is better to shift to a longer wavelength”. On the contrary, the necessity of high wavelengths emissions is well explained in the next paragraph.

Answer: We thank the reviewer for his comment. The following information was added to the Doped CDs section (section 4.3) of the manuscript (lines 307 on page 9).

Now we have justified the sentence by adding this higher wavelength has higher penetration depth and is harmless to tissues.

The authors should illustrate the sensing mechanism in the first paragraphs of this review, in order to make easier the readability.

Answer: We thank the reviewer for his comment, and we are happy to discuss it. We have illustrated the sensing mechanism in the manuscript.  It is in section 1 starting from line 51 on page 2.

The reference to the pandemic problem of Covid-19 in the conclusions deserves more space in the review.

Answer: We thank the reviewer for his comment, and we are happy to add this. The following information was added to the conclusion section (section 5) of the manuscript (lines 395 on page 11).

We have added more explanation and a recent reference. There are some reports available that discuss one of the strategies to fight the outbreak of COVID 19 is the real-time monitoring of proteolytic activity with potential protease inhibitors as biosensors for COVID 19. In one of the reports COVID 19 genome was used as the input for an AND logic function. Two types of fragments were used as inputs. This detection was based on exonuclease III and DNAzyme.

Reviewer 2 Report

In this review article, authors describe the use of carbon dots in logic gate operations. They divided the discussed literature according to the composition of the carbon dots and then accordiong to the different type of logic gates. In my opinion, this is a review that was missing from literature, since, to the best of my knowledge, there is no similar review article, and this is an asset for this article. The articles discussed in this review seem to cover a broad spectrum of the published reports and many aspects have been taken into account. However there are some points that in my opinion need to be addressed to improve the manuscript.

First of all, the language throughout the text need improvement.

First of all, the abstract seems poor and does not highlight the multiple data presented in the manuscript. The same goes for the introduction section, where there is only one sentence at the end (line 58) about the scope. The scope must be more clearly set and presented, so that it will be better for readers.

In the introduction, the paragraph about the logic gates, must contain more information, such as what logic gates are, where they are used, what are the different types of logic gates, and provide examples and uses for some of them (for example the AND, NOT, OR etc.). It will make it easier for readers to further understand the content of the review article.

In line 80, there is a problem with the numbers of the section. Please correct the numbering throughout the text.

As regards the in-text references of previous works, in my opinion only the surname of the author should be mentioned in the text (for instance in line 87 instead of X-Hu only Hu, line 117 instead of C. Dong only Dong etc.).

Section 5. I believe that maybe this section should be presented first, so that readers will first understand the differences in the different types of logic gates output, followed by the mechanism and then the types of CDs. This is because in previous sections, authors mention different types of outputs. Without this section preceding it mayt be difficult for a reader to understand the text.

Finally, at some points in the text it would be better if authors commented on the reports they present or even compare them, instead of just presenting them. This would give much valuable information to the readers to understand the significant points from each study.  

Overall, I believe that this is a good review article with much potential, if the above points are corrected.

Author Response

Reviewers' comments

We would like to thank the reviewer for their detailed and constructive comments, which have greatly improved the manuscript.

Reviewer #2:

In this review article, the authors describe the use of carbon dots in logic gate operations. They divided the discussed literature according to the composition of the carbon dots and then according to the different types of logic gates. In my opinion, this is a review that was missing from the literature, since, to the best of my knowledge, there is no similar review article, and this is an asset for this article. The articles discussed in this review seem to cover a broad spectrum of the published reports and many aspects have been taken into account. However, there are some points that in my opinion need to be addressed to improve the manuscript.

First of all, the language throughout the text need improvement.

Answer: We thank the reviewer for his comment, and we are happy to improve it. Please find the new version.

First of all, the abstract seems poor and does not highlight the multiple data presented in the manuscript. The same goes for the introduction section, where there is only one sentence at the end (line 58) about the scope. The scope must be more clearly set and presented, so that it will be better for readers.

Answer: We thank the reviewer for his constructive comment. The following information was added to the Abstract of the manuscript (lines 13 on page 1).

In addition, we cover the advancement in CDs-based logic gates with the focus of understanding the fundamentals of how CDs have the potential for performing various logic functions depending upon their different categories. And in introduction section lines 84 page 3 we have added. It covers the latest advancement and development in the field of CDs-based logic gates, along with understanding the mechanism of CDs based logic systems. Here we have categorized logic output based on their received output as single, combinational, sequential and reversible. In a similar manner we have divided the CDs used for logic function depending upon their synthesis as pristine, functionalized, doped, co-doped and other CDs complexes.

In the introduction, the paragraph about the logic gates, must contain more information, such as what logic gates are, where they are used, what are the different types of logic gates, and provide examples and uses for some of them (for example the AND, NOT, OR etc.). It will make it easier for readers to further understand the content of the review article.

Answer: We thank the reviewer for his comment. The following information was added to the Introduction section (section 1) of the manuscript (lines 62-66 on page 2).

Logic gates are the elementary building blocks for any digital system, where there is a certain relationship between the input and output signal based on a certain logic. The seven basic logic gates include AND, NOT, NOR, XOR, NAND, and XNOR. Along with it, there are integrated logic gates such as INHIBIT and IMPLICATION circuits. These logic gates have several applications, such as AND gates are used for data transfer, while NAND gates are used in alarms and buzzers. They are used in circuits that involve processing and computation.

In line 80, there is a problem with the numbers of the section. Please correct the numbering throughout the text.

Answer: We thank the reviewer for his comment and apologize for the mistake and now it is corrected throughout the text.

As regards the in-text references of previous works, in my opinion only the surname of the author should be mentioned in the text (for instance in line 87 instead of X-Hu only Hu, line 117 instead of C. Dong only Dong etc.).

Answer: We thank the reviewer for his comment. We have corrected it throughout the text.

Section 5. I believe that maybe this section should be presented first, so that readers will first understand the differences in the different types of logic gates output, followed by the mechanism and then the types of CDs. This is because in previous sections, authors mention different types of outputs. Without this section preceding it may be difficult for a reader to understand the text.

Answer: We thank the reviewer for his comment. We have changed the sequence according to the reviewer's suggestion. Section 2. Logic output (line 88, page 3) followed by section 3 sensing mechanism (line 175, page 5), and then section 4 Carbon dots design for logic function (line 203, page 6).

Finally, at some points in the text it would be better if authors commented on the reports they present or even compare them, instead of just presenting them. This would give much valuable information to the readers to understand the significant points from each study. 

Answer: We thank the reviewer for his comment. The following information was added to the single output section (section 2.1) of the manuscript (lines 128 on page 4). “Apart from the fluorescence technique, using magnetic resonance signal for dual readout logic operations is of significant importance. The combination of FL/MR techniques gives the logic devices better applicability in case of biological application”. Next, we have added: “The supramolecular strategies serve as a substitute for covalent modification and simplify the fabrication process” to the Combinational logic output section (section 2.2) of the manuscript (lines 138 on page 4). And ‘Fluorescence techniques have certain limitations, such as shorter emission lifetime of nanoseconds leading to inner filter emission, overlapping of excitation and emission spectra, interference from the light scattering, and short-lived autofluorescence species which can lead to inner filter effect (IFE). These limitations can be overcome by the triplet excited state phenomena known as ‘Phosphorescence’ was added to the sequential logic output section (section 2.3) of the manuscript (lines 152 on page 4). “The phosphorescence logic gates are superior to most common fluorescence-based logic gates due to their benefits over fluorescence” to the sequential logic output section (section 2.3) of the manuscript (lines 156 on page 4). Lines such as ‘This kind of operative multiphase logic response are scare in literature, developing a solid phase integrated logic function could be a good option for the researchers’ is added to Pristine CDs (section 4.1) on page 6, line 231. ‘This study confirms that the functional groups present on CDs surfaces imparts certain association affinity of CDs towards certain metal ions’ is added to functionalized CDs (section 4.2) on page 8, line 280. ‘There are very few reports where absorption technique is used as an output signal for logic operation’ is added to doped CDs (section 4.3) on page 9, line 306. ‘In this work the waste cigarette butts were converted to fluorescent CDs by simple hydrothermal process without using any expensive instruments and having variety of application make this work interesting’ is added to co doped CDs (section 4.4) on page 10, lines 348.

Round 2

Reviewer 1 Report

The manuscript can be accepted in the present form.

Reviewer 2 Report

After addressing the issues i mentioned previously, i believe that the manuscript is improved ad thus, it can be considered for publication.